# A Systematic Review of Bone Bruise Patterns following Acute Anterior Cruciate Ligament Tears: Insights into the Mechanism of Injury

**DOI:** 10.3390/bioengineering11040396

**Published:** 2024-04-19

**Authors:** Sueen Sohn, Saad Mohammed AlShammari, Byung Jun Hwang, Man Soo Kim

**Affiliations:** 1Department of Orthopedic Surgery, Inje University Sanggye Paik Hospital, College of Medicine, Inje University, Seoul 01757, Republic of Korea; osdocsse@gmail.com; 2King Abdulaziz Air Base Hospital, Ministry of Defense, Dhahran 34641, Saudi Arabia; s3dmohammed90@gmail.com; 3Department of Orthopaedic Surgery, Seoul St. Mary’s Hospital, College of Medicine, The Catholic University of Korea, Seoul 06591, Republic of Korea; soybeankk@naver.com

**Keywords:** bone bruise, anterior cruciate ligament, injury, tear, mechanism, review

## Abstract

(1) Background: The purpose of this systematic review was to determine the prevalence of bone bruises in patients with anterior cruciate ligament (ACL) injuries and the location of the bruises relative to the tibia and femur. Understanding the relative positions of these bone bruises could enhance our comprehension of the knee loading patterns that occur during an ACL injury. (2) Methods: The MEDLINE, EMBASE, and the Cochrane Library databases were searched for studies that evaluated the presence of bone bruises following ACL injuries. Study selection, data extraction, and a systematic review were performed. (3) Results: Bone bruises were observed in 3207 cases (82.8%) at the lateral tibia plateau (LTP), 1608 cases (41.5%) at the medial tibia plateau (MTP), 2765 cases (71.4%) at the lateral femoral condyle (LFC), and 1257 cases (32.4%) at the medial femoral condyle (MFC). Of the 30 studies, 11 were able to assess the anterior to posterior direction. The posterior LTP and center LFC were the most common areas of bone bruises. Among the 30 studies, 14 documented bone bruises across all four sites (LTP, MTP, LFC, and MFC). The most common pattern was bone bruises appearing at the LTP and LFC. (4) Conclusions: The most frequently observed pattern of bone bruises was restricted to the lateral aspects of both the tibia and femur. In cases where bone bruises were present on both the lateral and medial sides, those on the lateral side exhibited greater severity. The positioning of bone bruises along the front–back axis indicated a forward shift of the tibia in relation to the femur during ACL injuries.

## 1. Introduction

The anterior cruciate ligament (ACL) is essential for knee stabilization, preventing the tibia from moving forward relative to the femur, especially during activities requiring abrupt stops and directional changes [1,2]. The ACL plays a pivotal role in knee stability, working in conjunction with the posterior cruciate ligament to form an X-shaped structure inside the knee [1,2]. This configuration is essential in limiting excessive the forward movement of the tibia and contributing to the knee’s rotational equilibrium, especially under side-to-side (varus or valgus) stresses [3,4,5,6]. An ACL tear is among the most frequently observed and severe injuries in sports [7,8]. ACL tears are among the most prevalent ligament injuries of the knee in the United States, with an estimated annual occurrence rate of 68.6 per 100,000 people [9]. In Norway, there were 34 ACL injuries per 100,000 individuals [10], while Denmark reported 38 ACL injuries per 100,000 people [11], and Sweden saw 32 ACL injuries per 100,000 people [12]. The number of ACL tears in South Korea is increasing [13]. The expense of surgical repair, the duration of recovery, and the potential for lasting impairment have sparked considerable interest in preventing ACL injuries [14,15,16]. To prevent these injuries, it is essential to understand the mechanisms behind ACL injuries and to identify their risk factors [17,18].

Bone bruising, also known as subchondral bone marrow edema or edema-like marrow signal intensity, is detected via MRI scans in more than 80% of acute ACL injury cases [19,20]. Such bruises are thought to emerge from inflammation, swelling, and tiny fractures within the bone’s trabeculae due to the compressive forces between the femur and tibia during injury [21]. These bruises on the tibiofemoral joint serve as a ‘footprint’ that reflects the dynamics within the joint at the moment of the ACL tear, offering clues to the movements leading to the injury [22]. Femoral and tibial bone bruises are believed to mark the point of impact at the moment of injury, providing clues on how an ACL injury occurs [23,24].

The literature frequently documents the pattern of these bruises in cases of ACL injuries, often associating them with a mix of anterior tibial displacement, valgus stress, and either an internal or external rotation of the tibia [25,26,27]. Earlier research showed that bone bruises associated with ACL injuries occur most frequently in the lateral compartment, suggesting that a valgus force is the main cause of such injuries [28]. Alternatively, some studies have suggested that the primary mechanism behind an ACL injury could be the anterior translation of the tibia with minimal bending of the knee, as inferred from bone bruises found on the back of the tibia and the front of the femur [26,29].

While numerous investigations have explored the locations of bone bruises on the femur and tibia in individuals with ACL tears, studies have typically addressed the prevalence and sites of bone bruises on the tibia and femur in isolation for patients with ACL injuries [26,29]. There is limited knowledge about the comparative locations of bone bruises on the tibia and femur [26,29]. Additionally, most studies only focus on the medial to lateral direction of the tibia and femur at the bone bruise location, and there is a lack of research on the anterior to posterior direction [27,30]. Understanding the relative positions of these bone bruises could enhance our comprehension of the loading patterns during an ACL injury [27]. The purpose of this systematic review was to determine the prevalence of bone bruises in patients with ACL injuries and the common knee loading patterns that occur during ACL injuries based on the location of the bone bruises on the tibia and femur.

## 2. Materials and Methods

This study was performed following the guidelines of the Preferred Reporting Items for Systematic Reviews and Meta-Analysis (PRISMA) statement (S1 PRISMA Checklist) [31].

### 2.1. Data and Literature Sources

This study was performed in accordance with the Cochrane Review’s methods. Multiple comprehensive databases (MEDLINE, EMBASE, and the Cochrane Library) were searched in January 2024 for studies in English that evaluated bone bruises following ACL injuries (S1 Search Strategy). The search terms were as follows: “(bone OR osseous) AND (bruise OR contusion OR lesion OR edema) AND (anterior cruciate ligament OR ACL)” (Figure 1). Following the initial electronic search, reference lists and bibliographies of the discovered articles, including pertinent reviews and meta-analyses, were manually searched to identify trials potentially overlooked in the electronic search. Each identified article was then evaluated individually for inclusion.

### 2.2. Study Selection

Study inclusion was independently assessed by two reviewers in accordance with the established selection criteria. Titles and abstracts were initially reviewed for relevance. When there was uncertainty, the complete article was examined to decide its eligibility. Any differences in opinion were settled through discussion. The articles were included based on the following criteria: the research must have involved more than 15 human participants with ACL injuries; utilized MRI technology to evaluate bone bruises; documented the location of these bruises in at least one of the following compartments: the medial or lateral compartment of the femur, specifically the medial femoral condyle (MFC) or lateral femoral condyle (LFC), and the medial or lateral compartment of the tibia, specifically the medial tibial plateau (MTP) or lateral tibial plateau (LTP); and provided details on the prevalence of such injuries. All ACL injuries were included regardless of the ACL injury mechanism. Additionally, only articles written in English and published between 2010 and 2023 were eligible. The exclusion criteria eliminated case studies, systematic reviews without original data, research that only indicated the maximum occurrence and prevalence of bone bruises without identifying the specific knee compartment, and studies that used cadaveric models to investigate ACL injuries.

### 2.3. Data Extraction

Two reviewers independently extracted data from each study using a standardized data extraction form. Disagreements were resolved via discussion, and those unresolved through discussion were reviewed by a third reviewer. The following variables were included: the first author, publication year, country, study type, timing of MRI relative to injury, MRI intensity, total ACL injury sample size, and bone bruise sample size. The bone bruise pattern was primarily characterized by distinguishing between the medial and lateral compartments of the tibia and femur, denoted as LTP, MTP, LFC, and MFC. Additionally, in cases where bone bruises could be identified from the anterior to the posterior direction, they were categorized as anterior, central, or posterior. Given that bone bruises can appear in one or multiple locations, the number of occurrence sites was recorded (from one to four areas, encompassing LTP, MTP, LFC, and MFC). We attempted to contact the study authors for supplementary information when there were insufficient or missing data in the articles. The third senior investigator was consulted to resolve any disagreement during data extraction.

## 3. Results

A study flow diagram showing the process for study identification, inclusion, and exclusion is provided (Figure 2). The initial electronic search yielded 1169 studies. Three additional publications were obtained through manual searching. In total, 104 potentially eligible studies were assessed for inclusion after screening the titles and abstracts. After we reviewed the full texts, an additional 42 studies were excluded, leaving 30 studies for the final analysis.

The study characteristics are summarized in Table 1. All studies were retrospective, and most were conducted in the United States or China. The shortest time from injury to MRI measurement was 3 weeks, while the longest was 90 days or 3 months. The total ACL sample size was 3872, of which 3288 cases had bone bruises (84.9%).

In the 30 studies included in this research, all recorded the prevalence of bone bruises in the LTP, MTP, LFC, and MFC. Among 3872 cases of ACL injury, bone bruises were observed in 3207 at the LTP (82.8%), 1608 at the MTP (41.5%), 2765 at the LFC (71.4%), and 1257 at the MFC (32.4%). The highest occurrence of bone bruises was noted at the LTP, while the lowest was at the MFC (Table 2). Of the 30 studies, 11 were able to assess the anterior to posterior direction. In total, 1115 bone bruises were reported to have occurred in the lateral compartment of the tibial plateau. Among these, 37 (3.3%) occurred in the anterior section, 136 (12.3%) in the central section, and 942 (84.4%) in the posterior section. Similarly, 568 bone bruises in the medial compartment of the tibial plateau were noted in the same set of studies, with 35 (6.1%) in the anterior, 56 (9.9%) in the central, and 477 (84.0%) in the posterior sections. Furthermore, 994 bone bruises in the LFC were documented in the selected studies, with 107 (10.8%) in the anterior, 844 (84.9%) in the central, and 43 (4.3%) in the posterior sections. Additionally, the MFC had 649 reported bone bruises, with 52 (8.0%) in the anterior, 520 (80.1%) in the central, and 77 (11.9%) in the posterior sections (Table 3).

Among the 30 studies, 14 documented bone bruises across all four sites (LTP, MTP, LFC, and MFC). The most commonly occurring event was bone bruising in two of the four sites (857 cases), followed by occurrences in three sites (439 cases), one site (382 cases), and all four sites (366 cases). Among cases where bone bruising appeared in only one site, there were 152 cases at the LTP, 144 cases at the LFC, 37 cases at the MTP, and 49 cases at the MFC. The most frequent occurrence of bone bruising in two sites was observed in the LTP and LFC (541 cases), followed by the MFC and MTP (129 cases) and the LTP and MTP (114 cases). When bone bruises were found in three sites, they occurred predominantly in the LTP, LFC, and MTP (340 cases), and in the LTP, LFC, and MFC (81 cases) (Table 4).

## 4. Discussion

The findings of this study offer insights into the locations of bone bruises, in particular, sections of the tibia and femur. Our data indicate that the most prevalent pattern of bone bruising was on the lateral aspects of both the femur and tibia, which could enhance our comprehension of the loading dynamics involved in ACL injuries.

Bone bruises were most common in the LTP and LFC, and when examined in the antero-posterior direction, the posterior LTP and center LFC showed the most frequent occurrence of bone bruises. This suggests that collisions between the LFC and the LTP are more common than those between the MFC and the MTP at the time of injury. The predominance of lateral compartment bruising over medial compartment bruising in ACL injuries observed in this study corroborates the results of earlier studies on ACL bone bruises [18,20,30,35,37,40,44,46,51,53,55]. The higher incidence of bone bruises in the lateral compartments could indicate the application of valgus force during an ACL injury, leading to an “opening” effect on the medial side. This observation aligns with prior research indicating that a valgus load is the main cause of ACL injuries [22,28,37].

The findings of this study indicate that an anterior translation of the tibia in relation to the femur occurs during ACL injuries, as evidenced by the frequency and placement of bone bruises. The data reveal that, in all the predominant bone bruise patterns identified in this study, bruises on the tibia were located in the posterior region, while those on the femur were found in the central region, aligning with previous studies that utilized MRI to examine bone bruise patterns [56]. These outcomes imply that, during injury, the central or front part of the femoral condyle likely made contact with the posterior part of the tibial plateau, suggesting significant anterior movement of the tibia relative to the femur during ACL injury [57,58]. For such high-energy contact to occur, the posterior part of the tibia must move forward relative to the femur on both sides. This pattern of bruising in the sagittal plane aligns with findings from previous studies that have documented the anterior translation of the tibia [18,26,48].

Furthermore, recent studies have recorded considerable anterior movement of the tibia in patients at the presumed moment of injury [22,59]. Given the ACL’s role as a primary barrier against anterior shear stress, significant anterior movement of the tibia could generate substantial anterior shear forces, leading to an ACL injury [27]. Thus, these combined findings indicate that anterior shear force in the sagittal plane may be a critical risk factor for ACL injuries [27]. 

The variation in bruising patterns along the sagittal plane of the MFC and LFC sheds light on the tibia’s rotation, either internally or externally, during injury [18,21,26,60]. While the exact position of the tibia relative to that of the femur (internally or externally rotated) at the time of injury cannot be directly deduced from the bone bruise locations, the uneven distribution in the front–back direction between the lateral and medial sides implies that there was rotation of the tibia around its long axis during the ACL injury [26,60]. Bone bruises in the anterior portion of the LFC are not significantly more frequently observed than are those in the anterior portion of the MFC, but they do occur more often. If the tibia were to translate anteriorly without any rotation, we would expect to see similar bruising patterns on the MFC and LFC [18,26]. However, anterior translation combined with internal rotation of the tibia leads to contact of the posterior region of the LTP with the LFC more anteriorly compared with the interaction between the MTP and the MFC. This is because the internal rotation of the tibia brings the posterior part of the LTP forward [18,21,26,60]. However, since the frequency difference is not substantial, it cannot be conclusively determined that the internal rotation of the tibia is one of the main mechanisms [61]. This study’s outcomes indicate that the pattern of ACL injury is intricate, involving a complex, multi-directional loading pattern rather than a simple, single-plane loading pattern [61].

Additionally, the bone bruise patterns highlighted in this review suggest knee hyperextension as another potential mechanism for non-contact ACL injuries. The data revealed that 3.5% of bone bruises in the LTP and 6.8% in the MTP were in the anterior regions, hinting at possible knee hyperextension in some ACL injury scenarios. It is plausible that some of the anterior femoral condyle bone bruises reported in the included studies could be attributed to this mechanism of injury [23,62]. There is relatively limited literature on hyperextension injuries associated with ACL injuries and bone bruise patterns [23,62]. However, several authors suggest that the typical pattern involves anterior tibial bruises, sometimes accompanied by anterior femoral bruises, which occur due to direct impact of the structures during the injury. These injuries may occur as a result of actions such as extreme force on the tibia with a planted foot or a forceful kick [23]. 

Despite the thorough comparisons and analyses, this study faced several constraints. First, a widespread issue with systematic reviews is that the caliber of the original data can restrict the overall quality of the research. All studies were retrospective. Hence, there is a need for more prospective research in this field. Second, the inclusion of only published data might introduce a reporting bias, given that negative outcomes are less frequently disclosed. Third, there is the issue of population diversity among the studies reviewed. Although all studies provided data on bone bruises in subjects with ACL injuries, many studies also included subjects who experienced concurrent injuries to ligaments and the meniscus. As a result, the ability of any given study to determine whether bone contusions are solely due to ACL injury or whether concurrent ligamentous and/or meniscus injuries may influence the injury pattern is limited. Fourth, only research results from 2010 onwards were included. It is believed that more appropriate results can be obtained if results prior to 2010 are included [21,27]. Finally, no classification was made regarding gender (men and women) [44], age (pediatrics and adults) [32,33], or injury type (contact or non-contact injury) [19]. Although these factors can have a significant impact on the pattern of bone bruising, they could not be distinguished clearly using the data included in this study, so they were expressed in an integrated manner.

## 5. Conclusions

The most frequently observed pattern of bone bruises was restricted to the lateral aspects of both the tibia and femur. In cases where bone bruises were present on both the lateral and medial sides, those on the lateral side exhibited greater severity. The positioning of bone bruises along the front–back axis indicated a forward shift of the tibia in relation to the position of the femur during an ACL injury. Knee valgus can occur during an ACL injury, yet the peak occurrence of knee valgus takes place following a significant forward movement of the tibia in comparison to the position of the femur, which is enough to result in an ACL injury.

## Figures and Tables

**Figure 1 bioengineering-11-00396-f001:**
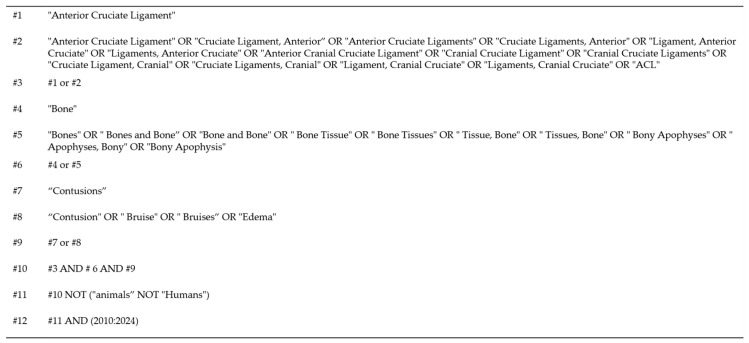
Search strategy for systematic review of bone bruise patterns following anterior cruciate ligament tears.

**Figure 2 bioengineering-11-00396-f002:**
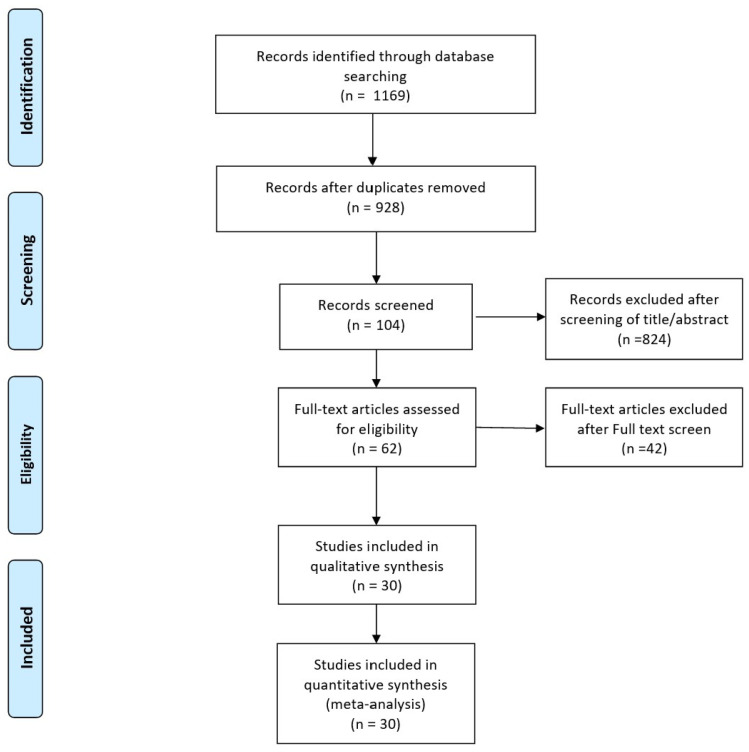
Flowchart illustrating the literature search process.

**Table 1 bioengineering-11-00396-t001:** Characteristics of included studies.

Author	Year	Nation	Period	Study Design	MRI Timing	MRI Intensity	Sample Size	Bone Bruise	Age	Gender (M)
Wang et al. [32]	2023	China	2011–2020	Retrospective	4 weeks	1.5 T	188	153	15.2	55
Vermeijden et al. [20]	2023	Netherlands	2019	Retrospective	31 days	1.5 or 3 T	254	251	34	114
Orringer et al. [33]	2023	USA	2015–2021	Retrospective	8 weeks		26 (Pediatric)	20	11.8	17
							26 (Adult)	20	34.3	17
Moran et al. [19]	2023	USA		Retrospective	30 days	3 T	78 (Contact)	75	23.1	54
							142 (noncontact)	138	25.6	77
Mester et al. [34]	2023	Germany	2016–2019	Retrospective	12 weeks		122	112	32.8	42
Li et al. [35]	2023	China	2021–2022	Retrospective	3 weeks	1.5 T	205	167	27.05	118
Galloway et al. [36]	2023	USA	2014–2019	Retrospective	2 months		60	57	23.4	31
Dai et al. [37]	2023	China	2020–2022	Retrospective	1 month	1.5 T	77	77	29.06	14
D‘Hooghe et al. [38]	2023	Italy	2014–2018	Retrospective			19	19	19.5	19
Brophy et al. [39]	2023	USA	2015–2019	Retrospective	3 weeks		191	181		
Moran et al. [6]	2022	USA	2018–2020	Retrospective	30 days	3 T	43	43	27.5	19
Kim et al. [40]	2022	Japan	2013–2021	Retrospective		2 T	176	141	26.8	98
Byrd et al. [41]	2022	USA	2015–2017	Retrospective	90 days	0.2–3 T	208	203	23.8	104
Agostinone et al. [42]	2022	Italy		Retrospective	3 months	1.5 T	29		29.1	24
Shi et al. [25]	2021	China	2016–2018	Retrospective	4 weeks	1.5 T	56 (Soccer)	43	30.3	2
							82 (Basketball)	66	26.7	6
Qiu et al. [43]	2021	China	2014–2020	Retrospective	1 month	1.5 T	138	93	32.06	29
Kim-Wang et al. [44]	2021	USA	2010–2013	Retrospective	6 weeks	1.5 T	136	135	64	54
Shi et al. [18]	2020	China	2016–2018	Retrospective	4 weeks	1.5 T	207	169	28.7	
Calvo et al. [45]	2019	Ireland	2014–2016	Retrospective	8 weeks	3 T	150	141	24.9	41
Bordoni et al. [46]	2019	Switzerland	2010–2018	Retrospective	90 days		78	54	14.3	41
Novaretti et al. [47]	2018	USA	2012–2016	Retrospective	6 weeks		53	51	13.3	26
Aravindh et al. [30]	2018	Singapore	2013–2016	Retrospective	6 weeks		168	155		126
Lattermann et al. [48]	2017	USA		Retrospective			81	81		
Berger et al. [49]	2017	Switzerland		Retrospective	8 weeks	1.5 T	107			
Song et al. [50]	2016	China	2011–2013	Retrospective	6 weeks	1.5 T	193		32.3	141
Filardo et al. [51]	2015	Italy	2004–2008	Retrospective	1 month		134	74	31.9	98
Witstein et al. [29]	2014	USA	2005–2010	Retrospective	6 weeks	1.5 T	73	70		28
Bisson et al. [52]	2013	USA	2005–2011	Retrospective	6 weeks	1.5 T	171	154	25.2	89
Yoon et al. [53]	2011	Korea	2006–2008	Retrospective	6 weeks		81	68	29	22
Jelic et al. [54]	2010	Serbia		Retrospective	1 month	0.3 T	120	39	31	88

**Table 2 bioengineering-11-00396-t002:** Bone bruise prevalence in medial and lateral compartments of femur and tibia.

Author	ACL Sample	Bone Bruise Sample	LTP	MTP	LFC	MFC
Wang et al. [32]	188	153	139	48	136	40
Vermeijden et al. [20]	254	251	240	32	163	138
Orringer et al. [33]	26	20	18	1	19	4
	26	20	16	9	10	12
Moran et al. [19]	78	75	70	47	65	49
	142	138	77	102	119	120
Mester et al. [34]	122	112	112	60	79	35
Li et al. [35]	205	137	167	90	135	62
Galloway et al. [36]	60	57	53	16	46	13
Dai et al. [37]	77	77	73	46	69	40
D’Hooghe et al. [38]	19	19	18	3	12	0
Brophy et al. [39]	191	181	154	93	140	44
Moran et al. [6]	43	43	35	28	35	27
Kim et al. [40]	176	141	82	47	116	29
Byrd et al. [41]	208	203	196	164	177	115
Agostinone et al. [42]	29	24	24	16	21	5
Shi et al. [55]	56	43	40	32	38	12
	82	66	62	31	42	20
Qiu et al. [43]	138	93	76	42	87	41
Kim-Wang et al. [44]	136	135	190	127	198	88
Shi et al. [18]	207	169	169	80	156	91
Calvo et al. [45]	150	141	141	89	131	59
Bordoni et al. [46]	78	54	44	11	57	34
Novaretti et al. [47]	53	51	51	37	51	20
Aravindh et al. [30]	168	155	141	95	132	50
Lattermann et al. [48]	81	81	76	46	66	20
Berger et al. [49]	107	96	96	42	44	5
Song et al. [50]	193	141	141	41	117	12
Filardo et al. [51]	134	74	35	11	23	5
Witstein et al. [29]	73	70	67	45	70	31
Bisson et al. [52]	171	154	145	44	132	11
Yoon et al. [53]	81	68	59	21	55	19
Jelic et al. [54]	120	39	20	12	24	6
Total	3872	3288	3207	1608	2765	1257

**Table 3 bioengineering-11-00396-t003:** Bone bruise prevalence in the anterior and posterior directions in the medial and lateral compartments of the femur and tibia.

Author	LTP Anterior	LTP Center	LTP Posterior	MTP Anterior	MTP Center	MTP Posterior	LFC Anterior	LFC Center	LFC Posterior	MFC Anterior	MFC Center	MFC Posterior
Vermeijden et al. [20]			235			30		153			104	
Moran et al. [19]	7	28	67	6	11	41	22	55	6	10	48	3
	9	42	46	18	28	94	42	93	5	22	114	55
D’Hooghe et al. [38]	1	1	16	0	0	3	1	11	0	0	0	0
Moran et al. [6]	1	7	27	3	4	21	0	35	0	4	22	1
Shi et al. [55]		2	38			32		38			12	
		1	61			31		42			20	
Qiu et al. [43]	4	5	67	2	1	39	6	78	3	0	38	3
Shi et al. [18]		8	161			80		156			88	3
Bordoni et al. [46]	9	16	34	2	4	9	18	41	20	10	25	12
Berger et al. [49]	6	26	64	4	6	33	5	31	8	2	3	0
Witstein et al. [29]			67			45		70			31	
Yoon et al. [53]			59		2	19	13	41	1	4	15	0
Total	37	136	942	35	56	477	107	844	43	52	520	77

**Table 4 bioengineering-11-00396-t004:** Summary of the distribution of bone bruises in the medial and lateral compartments of the femur and tibia.

Author	LTP only	LFC only	MTP only	MFC only	LTP + LFC	MFC + MTP	LTP + MTP	LTP + MFC	MTP + LFC	MFC + LFC	LTP + MTP + LFC	LTP + LFC + MFC	LTP + MTP + MFC	LFC + MTP + MFC	LTP + MTP + LFC + MFC	LTP + LFC + FH
Wang et al. [32]	13	13			47		4			1	25	20			19	11
Vermeijden et al. [20]	42				62		34				77				21	
Moran et al. [19]	6	1	16	18	64	31										
	8	50	8	26	69	94										
Li et al. [35]	32				34						39	11			62	
Dai et al. [37]	1				14	1	1	4	3		18	12	1		22	
D‘Hooghe et al. [38]	4				9		1				2				1	
Kim et al. [40]	8	38	3	0	29	1	11		8	9	15	10	2		7	
Byrd et al. [41]	4	0	4	1	22	0	11	2	1	0	46	10	4	1	97	
Shi et al. [55]					18						20				12	
					31						11				20	
Kim-Wang et al. [44]	6	2			29	0	5	2	1	2	33	7	1		47	
Shi et al. [18]	2				75			7	1		26	5	4		49	
Lattermann et al. [48]	5	6			23		4	2	1		22	6	3		9	
Witstein et al. [29]	14	21	2	2	10		43			29						
Jelic et al. [54]	7	13	4	2	5	2					6		2			
Total	152	144	37	49	541	129	114	17	15	41	340	81	17	1	366	11

## Data Availability

The data presented in this study are available in the main article.

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
