# Peer review of "A Systematic Review of Bone Bruise Patterns following Acute Anterior Cruciate Ligament Tears: Insights into the Mechanism of Injury"

_bioengineering, 2024, doi:10.3390/bioengineering11040396_

Round 1

Reviewer 1 Report

Comments and Suggestions for Authors

General information about article:

The article A Systematic Review of Bone Bruise Patterns Following Acute Anterior Cruciate Ligament Tears: Insights into the Mechanism of Injury presents a thorough analysis of bone bruises associated with ACL injuries, focusing on their prevalence and locations on the tibia and femur. By reviewing data from MEDLINE, EMBASE, and the Cochrane Library, the study finds that bone bruises are predominantly located at the lateral tibia plateau (LTP) and lateral femoral condyle (LFC), indicating a lateral stress mechanism during the injury. The analysis reveals that the most common bruising pattern suggests a forward shift of the tibia relative to the femur during ACL tears, offering crucial insights into the knee loading patterns involved in such injuries.

The study's conclusion—that bone bruises are most frequently observed on the lateral aspects of the tibia and femur, with lateral side bruises showing greater severity—provides important implications for the diagnosis, treatment, and prevention of ACL injuries. While the review is informative, it could be further enriched by discussing the limitations related to imaging techniques and the potential clinical implications of these findings for patient care and rehabilitation strategies.

Overall, this systematic review significantly contributes to our biomechanical understanding of ACL injuries and emphasizes the need for future research to explore the clinical applications of these insights in ACL injury management.

The work is generally well-written however, in places it will require minor changes and additions. Below are my detailed comments.

Minor comments:

Please expand the first paragraph with information on the role of the ACL in the biomechanics of the knee. Please add global injury information. Relevant information can be found in the works:

https://doi.org/10.3390/app9194102

References to literature in the text should come before the period. Please adapt the entire paper to the editorial requirements of the journal.

A figure depicting the selection of articles considered further in the study would be an excellent addition. Please consider adding one in the methodology section of the paper.

Please remove the redundant spacing common in the paper and bring the entire article in line with the requirements of the journal.

After making appropriate corrections to the content and literature, the work can be accepted for publication. Congratulates the authors on an interesting article and wishes them further success.

Author Response

General information about article:

The article A Systematic Review of Bone Bruise Patterns Following Acute Anterior Cruciate Ligament Tears: Insights into the Mechanism of Injury presents a thorough analysis of bone bruises associated with ACL injuries, focusing on their prevalence and locations on the tibia and femur. By reviewing data from MEDLINE, EMBASE, and the Cochrane Library, the study finds that bone bruises are predominantly located at the lateral tibia plateau (LTP) and lateral femoral condyle (LFC), indicating a lateral stress mechanism during the injury. The analysis reveals that the most common bruising pattern suggests a forward shift of the tibia relative to the femur during ACL tears, offering crucial insights into the knee loading patterns involved in such injuries.

▶We thank the reviewer for his/her valuable time and we agree with this succinct summary of our study.

The study's conclusion—that bone bruises are most frequently observed on the lateral aspects of the tibia and femur, with lateral side bruises showing greater severity—provides important implications for the diagnosis, treatment, and prevention of ACL injuries. While the review is informative, it could be further enriched by discussing the limitations related to imaging techniques and the potential clinical implications of these findings for patient care and rehabilitation strategies.

▶We thank the reviewer for his/her valuable time and we agree with this succinct summary of our study.

Overall, this systematic review significantly contributes to our biomechanical understanding of ACL injuries and emphasizes the need for future research to explore the clinical applications of these insights in ACL injury management.

▶We thank the reviewer for his/her valuable time and we agree with this succinct summary of our study.

The work is generally well-written however, in places it will require minor changes and additions. Below are my detailed comments.

▶We thank the reviewer for his/her valuable time and we agree with this succinct summary of our study.

Minor comments:

Please expand the first paragraph with information on the role of the ACL in the biomechanics of the knee. Please add global injury information. Relevant information can be found in the works:

https://doi.org/10.3390/app9194102

 ▶Thank you for your comments. We added the role of the ACL in the biomechanics of the knee and global injury information by referring to the research presented by the reviewer in the revised manuscript. (Lines 33-36, 39-43)

References to literature in the text should come before the period. Please adapt the entire paper to the editorial requirements of the journal.

 ▶Thank you for your comments. We made changes as suggested by the reviewer.

A figure depicting the selection of articles considered further in the study would be an excellent addition. Please consider adding one in the methodology section of the paper.

 ▶Thank you for your comments. Additional figures explaining study selection have been added to the methodology section of the revised manuscript. (Figure 1)

Please remove the redundant spacing common in the paper and bring the entire article in line with the requirements of the journal.

 ▶Thank you for your comments. Duplicate spaces were removed from the revised manuscript and changes were made to the revised manuscript as suggested by the reviewer.

After making appropriate corrections to the content and literature, the work can be accepted for publication. Congratulates the authors on an interesting article and wishes them further success.

 ▶Thank you for your comments.

Reviewer 2 Report

Comments and Suggestions for Authors

General

The current manuscript looks to provide a review of the current literature in regards to bone bruise patterns during ACL injury. The study used 30 articles and found that most bone bruises were on the lateral side of both the tibial plateau and femoral condyle. Interestingly, the anterior-posterior direction of bone bruises indicated a shearing force is likely during ACL injury. While this study is interesting and adds value to the current literature, some corrections will need to be made prior to publication.

COMMENTS

Methods

Your methods are well written and easy to follow. I believe there are a couple additions that need to be added for clarification.

1.      You need to include that your inclusion criteria were all studies regardless of mechanism of injury. While you mention this in the limitations, I found myself questioning whether these were isolated or complex ACL injuries while reading the manuscript.

2.      You need to explain why you had a cut off date of 2010. Why were articles prior to 2010 not included?

Results

Page 6, Line 134: You need to include the percentage (%) for MTP bone bruises.

Table 1, 2, and 3: Why are Orringer et. al. (28), Moran et. al. (14), and Shi et. al. (51) separated into 2 different rows? This needs to be explained somewhere.

Discussion

Page 9, Lines 170-171: Your sentence that begins “Upon analyzing the locations…” can be deleted. It’s basically repeating the same information from above.

Author Response

General

The current manuscript looks to provide a review of the current literature in regards to bone bruise patterns during ACL injury. The study used 30 articles and found that most bone bruises were on the lateral side of both the tibial plateau and femoral condyle. Interestingly, the anterior-posterior direction of bone bruises indicated a shearing force is likely during ACL injury. While this study is interesting and adds value to the current literature, some corrections will need to be made prior to publication.

▶We thank the reviewer for his/her valuable time and we agree with this succinct summary of our study.

COMMENTS

Methods

Your methods are well written and easy to follow. I believe there are a couple additions that need to be added for clarification.

You need to include that your inclusion criteria were all studies regardless of mechanism of injury. While you mention this in the limitations, I found myself questioning whether these were isolated or complex ACL injuries while reading the manuscript.

▶Thank you for your comments. As suggested by the reviewer, inclusion criteria were added to include all ACL injuries regardless of mechanism of injury. (Line 102)

You need to explain why you had a cut off date of 2010. Why were articles prior to 2010 not included?

▶Thank you for your comments. In the case of previous systematic review papers related to bone bruise in ACL injury, most of the included studies were studies before 2010, so the bone bruise pattern and injury mechanism in ACL injury were analyzed in studies after 2010 that are distinct from previous studies. We decided that it would be appropriate to judge this and made this setting. However, this can be considered a limitation because it excludes data that could have led to more appropriate conclusions by including more research results, as suggested by the reviewer. We added this issue in the limitation of revised manuscript. (Lines 232-234)

Results

Page 6, Line 134: You need to include the percentage (%) for MTP bone bruises.

▶Thank you for your comments. We added this issue in the revised manuscript. (Lines 138-139)

Table 1, 2, and 3: Why are Orringer et. al. (28), Moran et. al. (14), and Shi et. al. (51) separated into 2 different rows? This needs to be explained somewhere.

▶Thank you for your comments. We added this issue in the revised manuscript. Orringer's study distinguished between children and adults, Moran's study distinguished between contact and non-contact injuries and Shi's study distinguished between soccer and basketball, so they were marked separately in the table. According to the reviewer's opinion, the relevant information has been added to Table 1 of the revised manuscript. (Table 1)

Discussion

Page 9, Lines 170-171: Your sentence that begins “Upon analyzing the locations…” can be deleted. It’s basically repeating the same information from above.

▶Thank you for your comments. We deleted it in the revised manuscript. (Line 174)